# 3D Co-Printing and Substrate Geometry Influence the Differentiation of C2C12 Skeletal Myoblasts

**DOI:** 10.3390/gels9070595

**Published:** 2023-07-24

**Authors:** Giada Loi, Franca Scocozza, Flaminia Aliberti, Lorenza Rinvenuto, Gianluca Cidonio, Nicola Marchesi, Laura Benedetti, Gabriele Ceccarelli, Michele Conti

**Affiliations:** 1Department of Civil Engineering and Architecture, University of Pavia, Via Adolfo Ferrata 3, 27100 Pavia, Italy; giada.loi01@universitadipavia.it (G.L.); franca.scocozza@unipv.it (F.S.); 2Human Anatomy Unit, Department of Public Health, Experimental and Forensic Medicine, University of Pavia, Via Forlanini 2, 27100 Pavia, Italy; flaminia.aliberti01@universitadipavia.it (F.A.); lorenza.rinvenuto01@universitadipavia.it (L.R.); nicola.marchesi01@universitadipavia.it (N.M.); laura.benedetti@unipv.it (L.B.); gabriele.ceccarelli@unipv.it (G.C.); 3Fondazione IRCCS Policlinico San Matteo, Center for Inherited Cardiovascular Diseases, Transplant Research Area, 27100 Pavia, Italy; 4Center for Life Nano- & Neuro-Science (CLN2S), Fondazione Istituto Italiano di Tecnologia, 00161 Rome, Italy; gianluca.cidonio@iit.it

**Keywords:** co-printing, murine myoblasts (C2C12), bioprinting, fibrinogen-based hydrogel, polycaprolactone

## Abstract

Cells are influenced by several biomechanical aspects of their microenvironment, such as substrate geometry. According to the literature, substrate geometry influences the behavior of muscle cells; in particular, the curvature feature improves cell proliferation. However, the effect of substrate geometry on the myogenic differentiation process is not clear and needs to be further investigated. Here, we show that the 3D co-printing technique allows the realization of substrates. To test the influence of the co-printing technique on cellular behavior, we realized linear polycaprolactone substrates with channels in which a fibrinogen-based hydrogel loaded with C2C12 cells was deposited. Cell viability and differentiation were investigated up to 21 days in culture. The results suggest that this technology significantly improves the differentiation at 14 days. Therefore, we investigate the substrate geometry influence by comparing three different co-printed geometries—linear, circular, and hybrid structures (linear and circular features combined). Based on our results, all structures exhibit optimal cell viability (>94%), but the linear pattern allows to increase the in vitro cell differentiation, in particular after 14 days of culture. This study proposes an endorsed approach for creating artificial muscles for future skeletal muscle tissue engineering applications.

## 1. Introduction

Cells are subjected to and influenced by various biomechanical stimuli present in their physiological microenvironment [1]: several studies have shown that localized stresses, such as the stiffness [2,3,4,5,6,7], micropatterning [8,9,10,11], porosity [12,13], and geometry of the substrate [14,15,16], strongly influence cell proliferation and differentiation.

However, most of these studies have used 2D manufacturing techniques, such as microcontact printing, which do not allow us to mimic the real complexity of in vivo tissues impairing thus the capability to predict the actual cellular response [17]. In this context, techniques such as bioprinting (BioP) allow the production of in vitro 3D cellular microenvironments by depositing bioinks (i.e., a mixture of a biomaterial (usually hydrogel) and biological components (such as cells) [18]) in a spatially controlled way. In our previous study [19], we used BioP to recreate an in vitro muscle fiber model obtaining excellent cell differentiation, especially on the structure edges; however, the interplay between the bioprinted model and the substrate was not investigated, although it is known that this may have a crucial role on myoblast behavior [20,21,22,23,24,25,26,27,28]. In particular, the curvature feature, i.e., the use of circular, angled substrates with portions characterized by a certain radius of curvature, has been shown by several authors to improve cell proliferation [21,22,23]. Unfortunately, the effect of the geometrical curvatures on the process of myogenic differentiation is unclear and needs to be further elucidated. Therefore, finding the optimal substrate geometry that can improve myotube differentiation, alignment, and function is critical to engineering muscle tissues [29].

Among the manufacturing techniques suitable for this purpose, 3D co-printing emerges, which is a BioP technique that allows the simultaneous deposition of two materials of different natures, e.g., a thermoplastic material and a hydrogel [30,31], allowing us to obtain multi-material 3D constructs [32,33,34,35].

In this regard, the application of co-printing in the field of muscle regenerative medicine is already reported in the literature; in these studies, the co-printed engineered muscles showed high cell viability, good in vitro differentiation [36,37], and high in vivo regenerative effect [37,38]. However, none of these studies has yet investigated the possibility of making supports using this technique and of studying the influence of the support geometry and its biomechanical consequences on muscle cell differentiation.

Based on these considerations, the present study proposes the application of the co-printing technique to realize supports and understand how geometric cues maximize the differentiation of C2C12 murine myoblast cells. First, we evaluated the capability of the co-printing technique to realize substrates for cell-loaded materials by fabricating linear supports with a central channel made of polycaprolactone (PCL) and containing a fibrinogen-based hydrogel loaded with C2C12 cells that were previously selected [19]. Biological tests were carried out to evaluate cell viability and differentiation. Then, once the biocompatibility of the co-printing technique is verified, we moved on to its application to evaluate the substrate geometry influence on cell differentiation. To this aim, three different geometries, namely linear, circular with different outer diameters (OD), and hybrid structures (combining linear and circular features), were created.

## 2. Results and Discussion

### 2.1. Linear Structures: Bioprinted vs. Co-Printed

#### 2.1.1. Gene Expression Analysis by Quantitative Real-Time PCR

As a preliminary test, the effect of co-printing on cell differentiation was tested. In this regard, we performed gene expression analyses comparing structures with the same geometry (linear) realized with different techniques, i.e., BioP and co-printing. Data related to bioprinted structures were derived from our previous study [19].

Gene expression analyses were performed at 7, 14, and 21 days of culture in differentiative conditions (Figure 1). The bioprinted constructs are named F, which stands for fibrinogen-based hydrogel, while the co-printed structures are named F + PCL, as the same hydrogel has been deposited on a PCL support. RT-qPCR was used to detect expression levels of myogenic genes in 3D structures; in particular, since there are first preliminary tests, we only considered the two muscle master genes, i.e., MyoD and MCK, normalized by the housekeeping PGK gene (Appendix A). At 7 days, the MCK gene was significantly expressed by cells cultured in the co-printed linear structures (Figure 1a). At 14 days co-printing significantly improves the differentiation, showing high expression levels of both MyoD and MCK genes. At 21 days, no statistically significant differences in MCK expression were found; however, MyoD was differentially expressed in the bioprinted structures, showing a still early differentiation. Overall, at 21 days, MyoD gene expression decreased significantly; as expected, the expression of this gene is related to early myogenic differentiation, while, after 21 days of culture, the decreases are in favor of the expression of late genes involved in the myogenic commitment and cell elongation. These data show that the co-printing process significantly improves cell differentiation, in particular at 14 days. For this reason, all the 3D structures that will be examined below in the discussion have been created through 3D co-printing.

### 2.2. Curvature Feature: Circular and Hybrid Structures

#### Gene Expression Analysis by Quantitative Real-Time PCR

To evaluate the effect of the curvature feature on cell differentiation, we performed gene expression analyses comparing the circular and serpentine-like structures realized by exploiting the co-printing technique. Analyses were performed at 7, 14, and 21 days of culture in differentiative conditions (Figure 2). Since, as mentioned in the introduction, the curvature feature has already been examined from the point of view of proliferation and not of differentiation, we decided to perform an in-depth differentiation analysis. For this reason, the expression levels of myogenic genes (i.e., MyoD, Myf5, Cyclin D1, Myh1, MCK, MCad, MyoG) in the 3D structures were detected by RT-qPCR normalized by the PGK gene.

At 7 days, no statistically significant differences are observed in the expression levels of the Cyclin D1 gene; all structures provide the same proliferative stimulus. The MyoD gene is differentially expressed in the serpentine-like structure compared to the circular ones. Otherwise, the Myf5 gene is differentially expressed in circular structures compared to the serpentine-like one.

At 14 days, as expected, Cyclin D1 expression starts to decrease for all structures; the proliferation process seems to reduce, and the differentiation increases. However, Cyclin D1 remains differentially expressed in the hybrid structure compared to the circular ones; so, this geometry promotes a proliferative process prolonged over time. The MyoG gene is differentially expressed in the circular structure with a small radius compared to the other cases examined, while the MyoD gene is differentially expressed in the serpentine-like structure.

Finally, at 21 days of culture, the Myh1 gene is differentially expressed in the circular structure with a small radius and in the hybrid structure compared to that with a large radius. The MCad and MCK genes are differentially expressed in the serpentine-like structure compared to the other analyzed cases. No statistically significant differences are observed for the MyoG, Cyclin D1, and Dysf genes (data not shown).

In conclusion, regardless of geometry at 7 days, as expected, proliferation (indicated by cyclin D1 gene expression) is elevated and decreases with increasing culture period in favor of the differentiation process. At 14 days (short-term differentiation), the expression of cyclin D1 is still high in the serpentine-like constructs; instead, in the circular structures, it decreases favoring the expression of other differentiation genes. Finally, at 21 days (long-term differentiation), among the different geometries analyzed, at the same proliferation rate (i.e., same reduced expression of cyclin D1), the serpentine geometry is the one that showed greater differentiation and high expression of MCK and MCad genes.

### 2.3. Comparison of All Tested Geometries

Finally, to evaluate the effect of substrate geometry on cell behavior, we performed viability and gene expression analyses comparing all the considered geometries.

#### 2.3.1. Live/Dead Staining

Live/dead staining was performed at different time points during the culture (1, 7, 14, and 21 days) (Figure 3). For all geometries, at 24 h after printing, cells are of a rounded shape and homogeneously distributed throughout the construct. Furthermore, at all time points and in all cases considered, cells demonstrated very high viability (>94%).

On day 7, C2C12 cells remained mainly round-shaped in the center of the construct without merging to form myotubes. However, especially in linear constructs, an initial differentiation of C2C12 began at the edges of the 3D constructs, where cell elongation appeared. This is probably due to an inhomogeneous diffusion of the crosslinking solution or to lower oxygen and nutrient concentrations within the 3D constructs (Figure 3b).

Finally, at 14 and 21 days, C2C12 cells grown in linear and serpentine-like structures merged forming primordial myotubes even in the most central part of the 3D structure, and the alignment was promoted by the linear areas of the printed constructs (Figure 3d,h).

As for C2C12 cells grown in circular structures, at 7 and 14 days of culture, cells retain a rounded-shaped and slowly begin to elongate only at day 21, especially at the edges of the constructs (Figure 3l,p).

The elongation of C2C12 cells significantly decreased when printed in circular 3D structures (Figure 3i–p); otherwise, the linear structure appeared to significantly increase myotube formation, and the alignment of the myoblast is mainly located at the edges (Figure 3b–d).

Therefore, the live/dead results indicated that under differentiative conditions, the linear structure effectively induced the alignment of myoblasts in particular at the boundary of the structure with respect to the other geometries tested. An inhomogeneous diffusion of the crosslinker in the center of the construct and/or a lower diffusion of nutrients/oxygen could be responsible for insufficient cell elongation.

#### 2.3.2. Gene Expression Analysis by Quantitative Real-Time PCR

Gene expression levels of C2C12 cells cultured under differentiative conditions in the different geometries at 7, 14, and 21 days were compared to evaluate the influence of the geometrical factor on the differentiation rate (Figure 4). The expression levels of the MyoD and MCK genes in the 3D co-printed structures were detected by RT-qPCR and normalized by the PGK gene. MyoD was analyzed for 7 and 14 days of culture to characterize early muscle differentiation, while MCK was analyzed for 21 days of culture as it indicates cell differentiation and maturation of muscle tissue.

At 7 days, the MyoD gene is differentially expressed in the serpentine-like structure. But, at 14 days, the linear geometry allows bursting cell differentiation, which reaches a very high gene expression value compared to all the other geometries examined. At 21 days, MCK is differentially expressed in the serpentine-like structure compared to the other ones. Thus, we obtained the best gene expression results at the 14-day time point, especially with the linear geometry, while the serpentine-like construct allows an improved late differentiation at 21 days. In conclusion, because one of our goals is to obtain a mature construct in the least time possible, linear geometry was selected. Indeed, it proves to be the best 3D structure to promote myoblast alignment along the printed filament in a short-time culture (14 days) and with the highest gene expression level.

### 2.4. Discussion

Cells are typically affected by biomechanical aspects of their micro-environment, such as substrate geometry. The curvature feature improves cell proliferation; however, its effect on differentiation is not clear and needs to be explored further [28,39].

Given these premises, in this study, we used the co-printing technique to create substrates and study the effect of their geometries on the differentiation of C2C12 murine myoblasts. Three different geometries—linear, circular with different outer diameters (OD 10 mm, OD 5 mm), and hybrid structures (linear and circular features in a serpentine-like construct)—were created by combining fibrinogen-based hydrogel loaded with C2C12 and PCL.

The results suggest that the co-printing technology does not impair cell viability and, indeed, significantly improves the differentiation at 14 days. However, following the excellent results obtained at 14 days, a decrease in both MyoD and MCK expression was observed at the subsequent 21-day time point. Considering MyoD, its expression significantly decreased, as we expected, because it is considered the master gene of early myogenic differentiation. While, regarding the MCK, we hypothesize that, following the high expression peak at 14 days, at 21 days, the translation process is favored compared to the transcription process.

Furthermore, all structures exhibited optimal cell viability (>94%). The linear pattern showed the best results as it allowed to increase in vitro cell differentiation after 14 days of culture.

Concerning biological experiments, contrary to other studies in the literature [15], statistically significant differences were observed in the gene expression of circular structures with different ODs. Specifically, at 21 days, the small radius structure showed a statistically significant differentiation compared to the large radius one. However, our structures are much larger than those previously studied [15], and therefore the results might not be comparable.

Furthermore, the linear geometry, as expected, showed a high degree of alignment, perhaps because the linear structure mimics as much as possible the structure of the muscle fiber. In several studies, myotube alignment has been modulated and guided by surface topography [27,28,40,41,42]. Instead, in our study, the alignment of cells was achieved on different geometries by using co-printing to create channels on the PCL supports for guiding the cells in them.

The selection of the linear structure is in contrast with other studies [15] which instead identified the hybrid ones as the best geometries for the C2C12 cell line. The reasons may be that in our serpentine architecture the curvature angle is greater (90°) compared to that of previous studies (30°) [15], and in the same structure we have included a discrete number of curvatures (three instead of one). Hence, we have less alignment and much more influence on the circular pattern due to the greater angle and number of the curvatures.

Finally, the application of this method allowed us to overcome some of the limitations that emerged in our previous study in which a bioprinted fiber muscle model had produced promising results [19]. In fact, in this work, the co-printing technique allowed us to create complex geometry supports for the bioink containment, which reduced the cell invasion on the Petri dish and will allow us in the future to introduce a mechanical stimulus that could improve the homogeneity of differentiation.

### 2.5. Limitations

Despite the success of the proposed co-printing strategy and the excellent results obtained, some limitations emerged. In particular, one of the main limitations concerns the inhomogeneous cell differentiation, located at the structure edges, probably both because structure edges are the areas where the stiffness and the crosslinking effect are greater and because cells need oxygen/nutrients and so they move to peripheral areas where the oxygen/nutrients concentration is higher. In this regard, given the significant impact of biomaterial stiffness on cell differentiation, this aspect will certainly be considered and investigated in future studies. Furthermore, structures with narrower channels could be designed to favor the formation of syncytia between cells. However, the bioprinter and hydrogel resolutions do not allow us to apply this solution. Therefore, to improve the homogeneity of differentiation, one solution is to apply mechanical stimulation. Several examples of mechanical stimulation-based bioreactors to improve muscle tissue maturation are reported in the literature [43,44,45,46,47]. This strategy could also reduce tissue maturation times which currently take up to 21 days. However, the PCL material selected for the support fabrication, due to its stiffness, does not allow mechanical stimulation. Furthermore, being a thermoplastic material it requires low printing speeds and consequently very long manufacturing times. For this reason, a solution may be to change the support material, which will have to be more elastic, autoclavable, and biocompatible. Finally, regarding co-printed linear structures, we obtained optimal results at 14 days of culture, but at the 21-day time point we observed a significant decrease in both MyoD and MCK gene expression (Figure 1c). The decrease in MyoD was an expected result since it is a gene expressed in early differentiation, as mentioned in Section 2.1.1, while the decrease in MCK, we hypothesize, may be related to a more advanced differentiation that favors the translation process to that of transcription. Therefore, this aspect will have to be further investigated and clarified in future studies.

## 3. Conclusions

Co-printing can be used to fabricate in vitro 3D models of muscle tissue with different architectures. We conclude that geometrical cues influence the differentiation process of C2C12 myoblasts. We investigated the viability and differentiation of C2C12 myoblasts on three main geometries—a linear, a circular, and a hybrid pattern, which combined linear and circular features in one geometrical unit. Our results clearly showed that the linear pattern, compared to the other structures, emerged as the optimal geometry to maximize the differentiation of C2C12 myotubes. After 14 days in culture, C2C12 cells were able to fuse forming aligned myotubes, in particular in the structure’s edges, with high expression levels of specific skeletal muscle markers, such as MyoD.

Thanks to these findings, the results reported herein could have implications for improving skeletal muscle tissue engineering and the design of bioreactors for muscle–skeletal TE applications.

## 4. Materials and Methods

For the fabrication of skeletal muscle constructs, two commercial materials, i.e., a fibrinogen-based hydrogel (CELLINK^®^ FIBRIN, Cellink AB, Göteborg, Sweden) and PCL pellets, and the mouse cell line C2C12 were used. The bioink was prepared by mixing the hydrogel and C2C12 cells. For the printing process, an extrusion-based pneumatic bioprinter (INKREDIBLE+^®^, Cellink AB, Göteborg, Sweden) with two print heads (PHs) was adopted. Both PHs were used for PCL and bioink co-printing: the first one (PH1) for printing PCL and the second one (PH2) for printing bioink. In the printed structures, viability and differentiation of C2C12 cells were analyzed at different time points (1, 7, 14, and 21 days) using, respectively, morphological tests by live/dead staining and gene expression analysis (by Real-Time PCR).

### 4.1. Cell Culture

C2C12 myoblasts (ATCC, CRL-1772™, Manassas, VA, USA) were cultured in Dulbecco’s Modified Eagle’s Medium (DMEM) supplemented with 10% fetal bovine serum, 1% penicillin/streptomycin (Sigma Aldrich, Burlington, MA, USA), 1% glutamine, and 2% sodium pyruvate at 37 °C in a humidified 5% CO_2_ atmosphere. At about 80% of confluence, cells were used for the experiments. The cell concentration in the bioink was approximately 25 × 10^6^ cells/mL, as previously selected [19]. Cell counting was performed using a Burker’s chamber and an Eclipse TE200 microscope (Nikon, Minato, Tokyo, Japan).

### 4.2. Biomaterials and Crosslinker

Two commercial materials were used, a fibrinogen-based hydrogel (CELLINK^®^ FIBRIN) and PCL pellets. The fibrinogen component of the selected hydrogel shown in our previous study was used to recreate a suitable microenvironment for the regeneration of muscle tissue in vitro [19]. Once printed, the bioink was ionically crosslinked using a calcium chloride (CaCl_2_)-based solution to develop a suitable structural integrity. Instead, PCL was selected because it is one of the widely used biomaterials in BioP due to its biocompatibility and mechanical strength. This polymer allows manufacturing supports with complex geometries for bioink containment in long-term cell cultures.

### 4.3. 3D Bioprinter

To co-print the selected materials the 3D bioprinter Cellink INKREDIBLE + (Cellink AB, Sweden) was employed. It is a pneumatic extrusion-based 3D bioprinter equipped with two PHs, a UV LED curing system (365 and 405 nm), and a high-efficiency particulate air (HEPA) filter. The PHs temperature can be set up to a maximum of 130 °C. The printing chamber can guarantee the sterility necessary for cell-type experiments through the activation of the 365 nm UV light, the positive-pressure airflow, and the H13 HEPA filter.

### 4.4. Co-Printing Process and 3D Constructs Culture

Before the co-printing process, the bioprinter was placed under a biological sterile hood. An aluminum cartridge with a 0.5 mm metal nozzle was used for 3D printing the PCL. In detail, the cartridge was filled with PCL pellets, inserted into the PH1 and heated to 120 °C for 30 min before printing to sterilize the material. UV light was turned on for 1 h to sterilize all the surfaces. The bioink was prepared by mixing the hydrogel with C2C12 cells (10:1 ratio). The bioink was transferred to a plastic cartridge, then a conical nozzle (0.41 mm inner diameter) was connected, and, finally, the cartridge was inserted into the PH2. The PH1 and PH2 printing temperatures were respectively kept constant at 90 °C and room temperature (RT). The XYZ axes were homed, the Z axis was calibrated, and the pressure and printing speed were set according to the material guidelines (10–15 kPa and 600 mm/min for the hydrogel, 300 kPa and 45 mm/min for the PCL). The parameters set for this experiment are summarized in Table 1. The process starts with a virtual CAD model that is translated into PH coordinates (the G-code) by slicing software. In particular, the 3D CAD model of structures was designed using the Autodesk Inventor^®^ software and then it was sliced using an open-source slicing software, Slic3r. The G-code was created and the 3D constructs were co-printed on a Petri dish. Then the constructs were crosslinked for 5 min at RT using a CaCl_2_-based solution, which covered the whole 3D structure. The crosslinking solution was subsequently removed from the constructs and DMEM culture medium was added. 3D co-printed constructs were incubated at 37 °C in a humidified 5% CO_2_ atmosphere and cultured for up to 21 days. The culture medium was refreshed every three days. Four days after co-printing, the differentiation process of C2C12-laden bioink was induced by using a differentiation medium composed of DMEM supplemented with 2% fetal bovine serum. Figure 5a shows a schematic representation of the 3D co-printing process described here.

### 4.5. 3D Structure

To study the substrate geometry effect on murine myoblast differentiation, three different structures were co-printed: linear, hybrid, and circular with different ODs. Figure 5b,c summarize the geometries designed and used for the 3D co-printing experiments. Linear and serpentine-like structure length was set at 20 mm, while the OD of circular structures was set at 10 and 5 mm. The channel width was 0.93 mm for all geometries. This channel thickness was the minimum distance we were able to achieve when printing PCL substrates (data not shown).

### 4.6. Live/Dead Staining

Live/dead staining provides a two-color fluorescence cell viability assay that is based on the simultaneous determination of live (green) and dead (red) cells with two probes. This study used calcein and ethidium homodimer (EthD-1), optimal dyes for this application. Calcein is well retained within live cells, producing an intense, uniform green fluorescence. EthD-1 enters cells with damaged membranes and undergoes a 40-fold fluorescence enhancement upon binding to nucleic acids, producing a bright red fluorescence in dead cells. Therefore, live/dead staining (Invitrogen) was used to assess and monitor cell viability throughout the biological experiment. For this reason, it was performed at four different time points (1, 7, 14, and 21 days of cell culture). According to the protocol, a solution was prepared consisting of 1.5 mL of phosphate-buffered saline (PBS), 3 μL of ethidium homodimer-1 (EthD-1), and 1.5 μL of calcein. Three-dimensional constructs were covered with 500 μL of this solution and incubated for 45 min in the dark, and then the solution was removed. The image acquisition was performed by a semi-confocal microscope (ViCo confocal, Nikon).

### 4.7. Total RNA Extraction and Quantitative Real-Time PCR

Quantitative real-time PCR (RT-qPCR) is a technique used to analyze gene expression by evaluating mRNA from samples. It is a widely used technique indicating cell gene profiling that varies during time in biological experiments. In this study, RT-qPCR was used to analyze the expression levels of myogenic genes in the 3D co-printed constructs. At the different time points (7, 14, and 21 days in culture), cellular genetic material was isolated from each sample using 300 μL of lysis buffer (TRIzol Reagent). Then, total RNA was extracted using the Directzol RNA Miniprep reagents and the manufacturer’s protocol (Zymo Research) and quantified by NanoDropTM (Thermo-Fisher Scientific). Reverse transcription of the cDNAs was performed using the iScript™ cDNA Synthesis Kit (Biorad). Then, to elucidate the C2C12 cell differentiation process, the gene expression of relevant myogenic differentiation markers was quantified. Specifically, different genes were selected according to the time points. MyoD, Myf5, and Cyclin D1 were analyzed at 7 days of culture, with MyoD and Myf5 as genes characterizing early muscle differentiation and Cyclin D1 for evaluating cell proliferation. At 14 days of culture, MyoD, MyoG, and Cyclin D1 were evaluated; MyoD was measured to study early differentiation, MyoG was measured for late differentiation, and, finally, Cyclin D1 was measured to evaluate cell proliferation. MCK, Myh1, and MCad were analyzed at 21 days of culture as genes indicating late differentiation and maturation of muscle tissue.

The reaction and data analysis were performed respectively by using Mini-Opticon Real-Time PCR System (BioRad Laboratories) and CFX Manager Software. The expression of each gene was studied in triplicate and normalized using the expression of a housekeeping gene, i.e., phosphoglycerate kinase (PGK). Relative gene expression was expressed in terms of fold increase calculated using the 2^(−ΔΔCt) method as described in this bibliography article [48]. The fold increase is a parameter that indicates how much a gene in the test condition considered is differently expressed compared to a control sample. The primer sequences used for gene expression analysis are listed in Appendix A.

## Figures and Tables

**Figure 1 gels-09-00595-f001:**
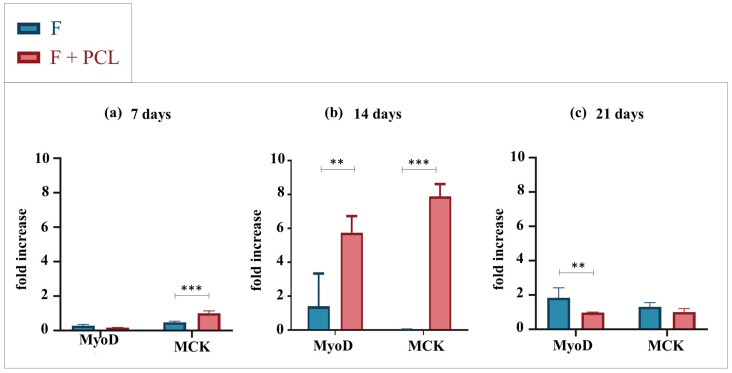
Gene expression analysis of C2C12 cells cultured into linear structures bioprinted (F) or co-printed (F+PCL) at 7, 14, and 21 days. (**a**) qRT-PCR at 7 days. (**b**) qRT-PCR at 14 days. (**c**) qRT-PCR at 21 days. Results are normalized to the housekeeping gene (3-phosphate dehydrogenase [PGK]). Statistically significant values are indicated as, ** *p* < 0.01 and *** *p* < 0.001. Analysis of variance test was performed to evaluate data significance.

**Figure 2 gels-09-00595-f002:**
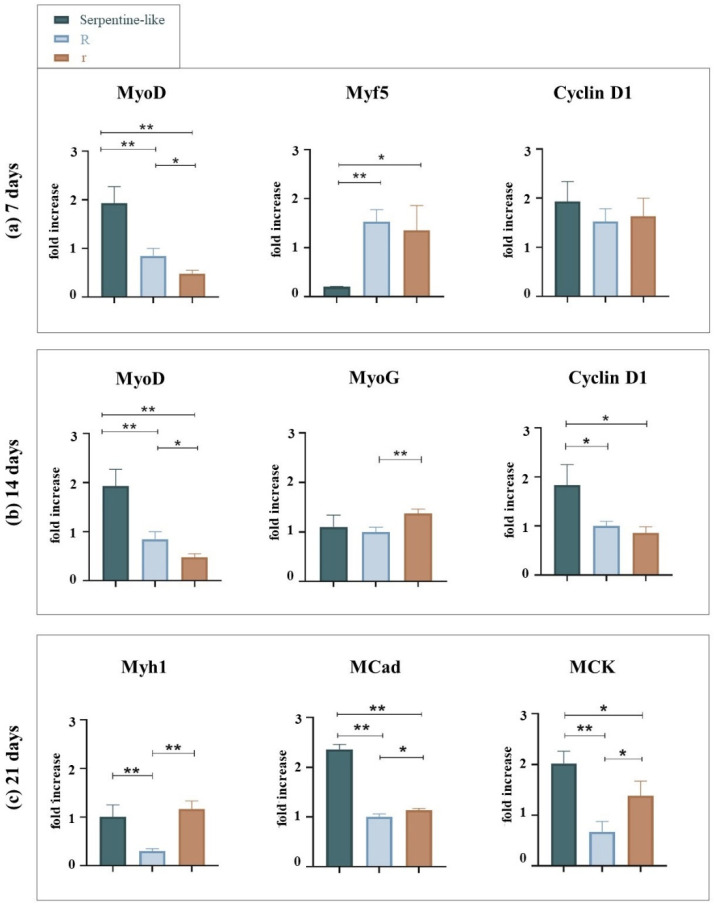
Gene expression analysis of C2C12 cultured into serpentine-like and circular (R, r) structures at 7, 14, and 21 days. (**a**) qRT-PCR at 7 days. (**b**) qRT-PCR at 14 days. (**c**) qRT-PCR at 21 days. Results are normalized to the housekeeping gene (3-phosphate dehydrogenase [PGK]). Statistically significant values are indicated as * *p* < 0.05, ** *p* < 0.01. Analysis of variance test was performed to evaluate data significance. R indicates structures with a 10 mm outer diameter while r indicates those with a 5 mm outer diameter.

**Figure 3 gels-09-00595-f003:**
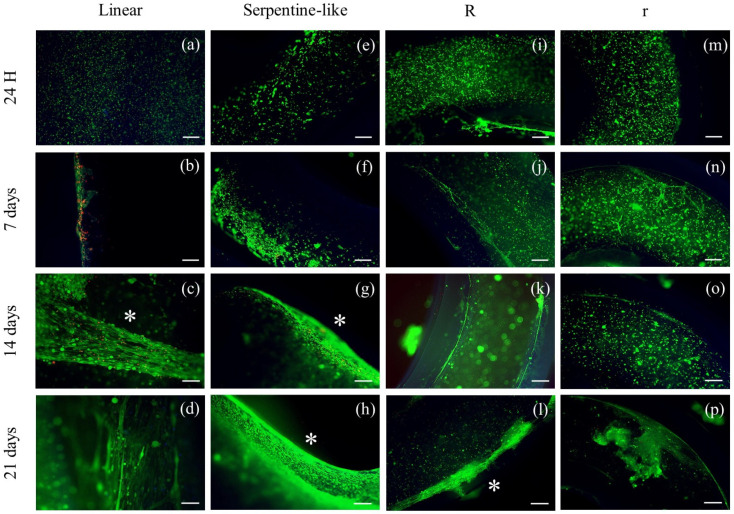
Live (green)/dead (red) images of different 3D geometries at specific time points in differentiative conditions. (**a**–**d**) Linear 3D constructs; (**e**–**h**) serpentine-like 3D constructs; (**i**–**l**) circular 3D constructs with OD 10 mm; (**m**–**p**) circular 3D constructs with OD 5 mm. Scale bar 50 μm. * highlights cell elongation. R indicates structures with a 10 mm outer diameter, while r indicates those with a 5 mm outer diameter.

**Figure 4 gels-09-00595-f004:**
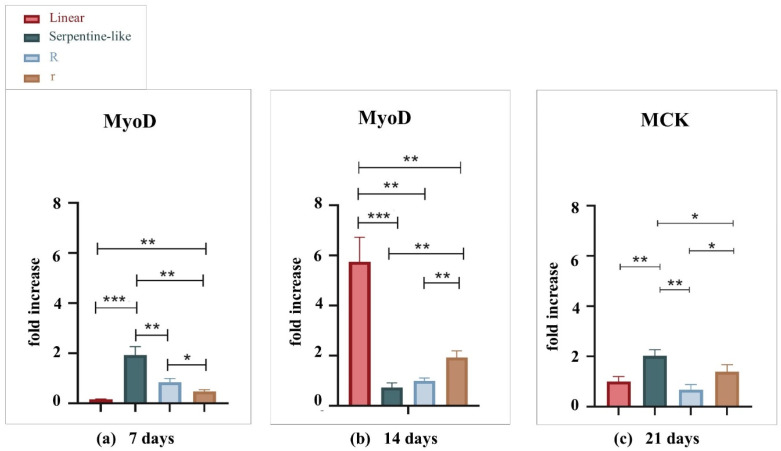
Gene expression analysis of C2C12 cultured into the different geometries tested (linear, serpentine-like, and circular) at 7, 14, and 21 days. (**a**) MyoD expression by qRT-PCR at 7 days. (**b**) MyoD expression by qRT-PCR at 14 days. (**c**) MCK expression by qRT-PCR at 21 days. Results are normalized to the housekeeping gene (3-phosphate dehydrogenase [PGK]). Statistically significant values are indicated as * *p* < 0.05, ** *p* < 0.01, and *** *p* < 0.001. Analysis of variance test was performed to evaluate data significance. R indicates structures with a 10 mm outer diameter while r indicates those with a 5 mm outer diameter.

**Figure 5 gels-09-00595-f005:**
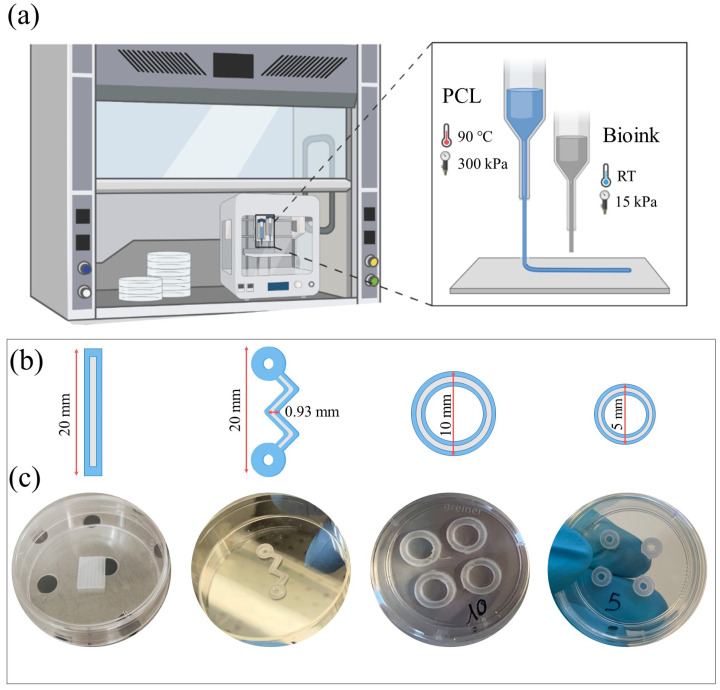
Design and fabrication of the different 3D constructs used in the study—linear, hybrid, and circular with different outer diameters (OD 10 mm, OD 5 mm). (**a**) Illustration showing the biofabrication process, selected materials, and printing parameters. (**b**) Schematic representation of the structures CAD design. (**c**) 3D-printed PCL constructs.

**Table 1 gels-09-00595-t001:** Summary of the operational parameters set for the study.

Operational Parameters	PCL	CELLINK FIBRIN
Extrusion pressure [kPa]	300	10–15
Conical nozzle diameter [mm]	0.5	0.41
Printing speed [mm/min]	45	600
Printing temperature [°C]	90	RT

## Data Availability

Data will be made available on request.

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
