# Peer review of "3D Co-Printing and Substrate Geometry Influence the Differentiation of C2C12 Skeletal Myoblasts"

_gels, 2023, doi:10.3390/gels9070595_

Round 1

Reviewer 1 Report

The aim of the study, specifically, the aspect of curvature and its effect on myogenic differentiation is not clear. The definition of curvature remains unclear throughout the paper.

English language and grammar needs further refinement.

Lines 322 state for PCL printing pressure was set to 300kPa and 7.5mm/min was the printing speed. According to table 1, printing parameters were 85kPa and 45mm/min.

Why are these value different? Or are these values based on manufacturer’s recommendations? In which case, it is advisable to change the language structure of the sentence.

Lines 94-98: “At 21 days, no statistically significant differences were found. These data show that the co-printing process significantly improves cell differentiation, in particular at 14 days. For this reason, all the 3D structures that will be examined below in the discussion have been created through 3D co-printing.”

Why was the result at day 21 not significant? From figure 1, the fold increase drastically decreases. Did the authors explore the reasons as to why this decrease occurs?

Lines 110-112: It is stated that “The expression levels of myogenic genes (i.e., MyoD, Myf5, Cyclin D1, Myh1, MCK, MCad, MyoG) in the 3D structures were detected by RT-qPCR normalized by the PGK gene.” on circular and serpentine printed structures, however in line 86-90, only MyoD and MCK expression levels were tested. Why were one set of patterns tested for fewer myogenic genes and the other tested for more? Why weren’t both patterns tested for all myogenic genes?

Lines 237-238: the term numerosity does not make sense. Kindly provide further explanation.

General Comments:

Porosity also plays a critical role in cellular differentiation, which has not been explored in the present study. Kindly have a look at the following publications. https://www.frontiersin.org/articles/10.3389/fbioe.2021.629270/full, https://www.ncbi.nlm.nih.gov/pmc/articles/PMC8203544/

Another aspect that has not been studied is the mechanical properties of both the PCL and Fibrin inks. While stiffness has been mentioned in the paper as an aspect, the current study has not measured what the final stiffness of the material is and whether that too plays a role in myogenic differentiation. It would be interesting to know if the composition of the bioinks can be tuned further to mimic in vivo mechanical properties in addition to in vivo geometries.

English needs improvement 

Author Response

We are grateful to the Reviewers for their valuable feedback. All their comments have been considered and, hopefully, addressed to create a revised version of the manuscript.

Accordingly, please find below a point-by-point rebuttal answering the Reviewers’ concerns in the attached file.

Reviewer 2 Report

The paper explores the influence of 3D co-printing and substrate geometry on the differentiation of C2C12 skeletal myoblasts. The authors argue that the biomechanical aspects of the cellular microenvironment, such as substrate geometry, significantly influence cells. They demonstrate that the 3D co-printing technique allows the creation of substrates and significantly improves differentiation at 14 days. The authors also investigate the influence of substrate geometry by comparing three different co-printed geometries – linear, circular, and hybrid structures (linear and circular features combined). The authors used two commercial materials, a fibrinogen-based hydrogel and PCL, and the cell line C2C12 for the fabrication of skeletal muscle constructs. The bioink was prepared by mixing the hydrogel and C2C12 cells.

The paper is well-structured and provides a comprehensive investigation of the influence of 3D co-printing and substrate geometry on the differentiation of C2C12 skeletal myoblasts. The authors' methodology is sound, and their findings contribute to the field of skeletal muscle tissue engineering. The paper could have implications for improving skeletal muscle tissue engineering and the design of bioreactors for muscle-skeletal Tissue Engineeing applications.

Critical points:

The authors have indeed highlighted an important parameter for cell growth and differentiation. However, the explanation of their experiment could be more accessible to those who are not specialists in tissue engineering. Given that the readership of this journal extends beyond the field of tissue engineering, it's crucial that the content is understandable to a wider audience. As someone who collaborates in this discipline but specializes in a different field, I found myself needing to read the paper multiple times to grasp the significance of these analyses.

Another point of critique is the use of the term "fold increase" in the axis description. The term appears to suggest a multiplication of an initial value, but this is not explicitly clarified. It would be beneficial to provide an explanation of this term either in the figure captions or within the main text to aid reader comprehension.

Author Response

We are grateful to the Reviewers for their valuable feedback. All their comments have been considered and, hopefully, addressed to create a revised version of the manuscript.

Accordingly, please find a point-by-point rebuttal answering the Reviewers’ concerns in the attached  file.
